# A Study of Strength Parameter Evolution and a Statistical Damage Constitutive Model of Cemented Sand and Gravel

**DOI:** 10.3390/ma16020542

**Published:** 2023-01-05

**Authors:** Honglei Ren, Xin Cai, Yingli Wu, Peiran Jing, Wanli Guo

**Affiliations:** 1Materials & Structure Engineering Department, Nanjing Hydraulic Research Institute, Nanjing 210029, China; 2College of Mechanics and Materials, Hohai University, Nanjing 211100, China; 3Geotechnical Engineering Department, Nanjing Hydraulic Research Institute, Nanjing 210029, China; 4State Key Laboratory of Water Resources and Hydropower Engineering Science, Wuhan University, Wuhan 430072, China; 5Dam Safety Management Department, Nanjing Hydraulic Research Institute, Nanjing 210029, China

**Keywords:** cemented sand and gravel (CSG), Mohr–Coulomb strength criterion, strength parameter, particle flow, acoustic emission, statistical damage

## Abstract

Cemented sand and gravel (CSG) has a wide range of applications in dam construction, and its properties are between rockfill and roller compacted concrete (RCC). A difference in gel content will result in a variance in CSG’s structure and mechanical properties. To investigate the intricate structural mechanical properties of CSG, this study conducted a series of laboratory tests and associated discrete element analyses. Accordingly, the evolution law of the strength parameters of CSG is explored and a statistical damage constitutive model suitable for CSG is established. The main contributions of this study are as follows: (1) The failure mechanism of the CSG was described from the microscopic level, and the evolution law of the strength parameter cohesion and friction angle of the CSG was analyzed and summarized. (2) Based on the particle flow model, the energy development law and the spatiotemporal distribution law of acoustic emission (AE) provide illustrations of the strain hardening–softening transition features and the interaction between cohesion and friction of CSG. (3) The evolution function between the strength parameter and the strain softening parameter was built, and the critical strain softening parameter was determined by the microcrack evolution law of the particle flow model. (4) The accuracy of the evolution curve was confirmed by comparing it to experimental results. (5) Based on the relationship between cohesion loss and material damage, a statistical damage constitutive model was developed using the improved Mohr–Coulomb strength criterion as the micro strength function. The constitutive model can accurately describe the stress–strain curves of CSG with different gel content. Furthermore, the model reflects the strain hardening–softening properties of CSG and reveals the relationship between the weakening of cohesion and material damage at the microscopic level. These findings provide valuable guidelines for investigating the damage laws and microcosmic failure features of CSG and other relevant materials.

## 1. Introduction

The cemented gravel dam was proposed in the 1970s [1], and this form of dam’s construction materials tend to be a small amount of cementing material mixed with pebbles taken from a construction site (or artificially broken aggregate). Thus, it is not necessary to build a particular yard [2]. The cemented gravel dam is characterized by easy construction, less environmental pollution [3,4], low hydration heat [5], excellent seismic performance, and strong adaptability to the foundation [6]. The structure of cemented sand and gravel (CSG) is complex and has the characteristics of both rock-fill and roller-compacted concrete (RCC). The mechanical behavior of CSG will significantly shift because of the change in gel content. Therefore, it is hard to find a classical strength theory or constitutive model to describe the complex mechanical properties of CSG [7].

Numerous laboratory physical tests have been employed for the mechanical properties of CSG. Lohani [8] thoroughly analyzed the influence of water content on the peak strength, peak strain, and Young’s modulus of CSG. Kongsukprasert [9,10] explored the effects of age, loading rate, and gel content on the mechanical properties of CSG, focusing on the nonlinear mechanical characteristic of CSG. Haeri [11,12,13] clarified that when no cemented material is added or the gel content of the total cemented material is less than 1.5%, the CSG will mainly show shear contraction, otherwise, the dilatancy will be distinct. In addition, the specimen illustrated obvious strain softening after the failure peak, and the cohesion, strength, and stiffness will enlarge with the increase of gel content. Zhang et al. [4] focused on the nonlinear mechanical behavior and damage characteristics of CSG under dynamic load. Most of the previous studies have focused on the description of the mechanical properties of the CSG while the mechanism of the mechanical properties has been less explored.

Many representative strength criteria in geotechnical engineering have been proposed, including the Tresca, Drucker–Prager, Mohr–Coulomb, and Hoek–Brown criteria [14]. According to these models, the strength of rock and soil mass mainly comes from the cohesion generated by the mutual attraction between particles and the friction generated by the relative motion between particles at the mesolevel. The friction angle φ and cohesion c are assumed to remain constant. However, it is hard for these traditional strength criteria to adequately describe the failure mechanism and damage to the evolution law of geotechnical materials. Thus, scholars examine the cohesion and friction angle of the strength parameters to revise these strength criteria. For example, Martin et al. [15,16] considered the damage of cohesion and friction angle; Hajiabdolmajid [17,18] and Edelbro C et al. [19] established a cohesion weakening and friction strengthening model. In addition, Pourhosseini et al. [20,21,22,23,24] created a constitutive model based on an improved strength criterion, which played a certain role in describing the mechanical properties of rock. Consequently, researchers can use the methods to develop a strength theory based on CSG material by considering the importance of taking the influence of gel content into account.

Classical constitutive models in geotechnical engineering include the Duncan–Chang nonlinear model [25], the Cambridge model [26], the Lade–Duncan elastoplastic model [27], and the generalized plastic model [28]. Under the influence of gel content, confining pressure, and aggregate gradation, the nonlinearity of CSG is distinct, and the elastoplastic mechanical behavior is evidently in the loading process. Many advisable explorations have been made on the constitutive model of CSG. Sun et al. [29] found a nonlinear elastic model based on the augmented spring method. Alan et al. [30] proposed an elastic-plastic yield model considering the viscosity effect. Lam et al. [31] created an elastoplastic constitutive model that takes into consideration the influence of gel content. Wu et al. [32] proposed an elastoplastic constitutive model based on the dual medium hypothesis. Nevertheless, a great number of test data demonstrate that, with an increase in gel content, the loading curve of CSG will show a distinct post-peak drop phenomenon [12,33,34,35], while the above-mentioned constitutive models find it hard to describe the underlying mechanical characteristics. Additionally, there is an internal connection between the characters of strain hardening–softening transformation and the evolution law of strength parameters.

Due to the CSG’s complicated structure and abundance of pores, it is challenging to find a suitable constitutive model to accurately depict the entire process of damage and failure. According to the statistical damage theory, the microscopic non-uniformity of materials leads to macroscopic damage, which results in a series of nonlinear mechanical behavior [36]. Therefore, the statistical damage constitutive model is appropriate to describe the damage characteristic of CSG. The key to establishing the statistical damage constitutive model is the selection of micro strength function [37,38]. Cao [39] and Zhu [40] used the Mohr–Coulomb criterion as the strength function to establish a statistical damage model. Chen [41] and Zhao [42] selected the Drucker–Prager strength criterion as the micro-strength function to create a statistical damage model for rock. Jiang et al. [43] and Cao et al. [44] established a statistical damage constitutive model that considers post-peak brittle drop and strain softening respectively. In addition, their statistical damage model is not only applicable to rock materials but also to concrete [45] and soil [46] materials.

Most previous studies have been conducted on complex mechanical properties of cemented sand and gravel, there are rarely strength theories and constitutive models that can systematically describe the damage mechanism of CSG. Furthermore, when the gel content is at a low level, the CSG material shows similar mechanical properties as rockfill material. It also has the same failure characteristics as rock and concrete material when its gel content is at a high level. Nevertheless, few previous studies have comprehensively considered these two characteristics. To this end, in order to narrow these research gaps, the primary objectives of this study could be listed as follows: (1) According to the triaxial shear test and particle flow model of CSG, this study aims to examine the strain-hardening and strain-softening properties of CSG and the underlying causes for the evolution of strength parameters. (2) The study also aims to analyze and establish the evolution of strength parameters c and φ along with strain softening parameters. (3) Taking the Mohr–Coulomb strength criterion and considering cohesion softening as the micro strength distribution function, the study aims to create a statistical damage constitutive model of CSG under different gel contents. This study further explores the micromechanical mechanism behind the complex mechanical properties of CSG materials and the damage characteristics of CSG under external load. The results of this study may provide valuable guidelines for the transformation of cemented gravel dams from a temporary project to a permanent assignment.

## 2. Mechanical Properties of CSG

### 2.1. Experimental Program

To investigate the influence of gel content on the mechanical properties of CSG, a series of large-scale triaxial shear tests were carried out. The gel contents were set as 20 kg/m^3^ (close to the mechanical properties of rockfill), 40 kg/m^3^, 60 kg/m^3^, 80 kg/m^3^, and 100 kg/m^3^ (close to the mechanical properties of RCC). Considering the working behavior and structural performance of low, middle, and 100-m highly cemented gravel dam, the confining pressure was set as 300 kPa, 600 kPa, 900 kPa, and 1200 kPa, respectively. The coarse and fine aggregates were the pebble material and medium coarse sand from the Nanjing suburb, and the specific aggregate gradation is shown in Table 1.

The cement material was P.O.42.5 ordinary Portland cement and fly ash (grades I) with a ratio of 1:1, and the water-binder ratio was 1.0. The mixture proportions of CSG are shown in Table 2. The specific test process is shown in Figure 1. The TYD-1500 static and dynamic triaxial test machine in the geotechnical Laboratory of Nanjing Hydraulic Research Institute was used in the test. The prepared specimen was coated with rubber film and hoisted to the triaxial test machine before loading. The axial pressure was applied by the transmission rod at the loading speed of 2 mm/min. When the stress started to stabilize, the test was terminated.

### 2.2. Strain Hardening–Softening Characteristics

Figure 2 shows the stress–strain curves of CSG under tri-axial shear tests with different gel contents. When the gel content is equal to 20 kg/m^3^, there is almost no strain-softening stage, and the stress–strain curve is close to that of rock-fill material. With the increase of gel content, the post-peak strain softening behavior becomes far more distinct. When the gel content is equal to 100 kg/m^3^, the curve shows a residual stage, and the failure mode of CSG approaches that of RCC material. The failure strength of CSG is composed of the bonding force between cementitious and gravel, the bite force between particles, and the friction resistance. When the gel content is at a low level, the interaction between aggregates is mainly spanning or overturning, and the increase of confining pressure will aggravate this phenomenon. The effect of bonding force between aggregates is implicit and the failure strength is mainly friction resistance, resulting in a strain softening phenomenon after the peak is no longer evident. With the increase of gel content, the bonding force between aggregates takes the dominant position. The shear failure of CSG occurs when the micro-cracks in the cement material are connected. The shear displacement overcomes the intergranular occlusion and the bonding force between the cement material and the gravel material. Subsequently, the loose particles of the gravel in the shear failure area increase and the cohesion *c* of the material decreases, showing the phenomenon of strain softening on the macro level. The friction between gravel and stone materials can also make the specimen bear a certain external load, which will generate residual strength. The strain hardening and softening characteristics of rock will directly impact the accuracy of numerical analysis results and the reliability of engineering safety evaluation [47]. As for the CSG material with more complex mechanical properties, it is essential to analyze the strain-hardening and strain-softening characteristics.

Younes Amini [33] studied the brittleness characteristics of CSG, and their results show that the increase of gel content significantly improves the brittleness degree of CSG, which was affected by the loading method. However, a quantitative analysis of the brittleness drop degree of CSG has not been conducted. The concept of brittleness index in rock is here introduced to further explore the post-peak strain hardening–softening transformation characteristics of CSG. The brittleness index represents the drop degree of the post-peak curve, which can reflect the strain-softening level of CSG. The brittleness index has a variety of expression forms, and two representative calculation methods were selected as follows:(1)$$B_{1}=\frac{\varepsilon_{r}\left(\sigma_{p}-\sigma_{r}\right)}{\sigma_{p}\left(\varepsilon_{r}-\varepsilon_{p}\right)}+\frac{2}{\pi \varepsilon_{p}}\arctan\frac{\sigma_{p}}{\varepsilon_{p}}$$
(2)$$B_{2}=\frac{\left(\sigma_{p}-\sigma_{r}\right)}{\left(\varepsilon_{r}-\varepsilon_{p}\right)}+\frac{\left(\sigma_{p}-\sigma_{r}\right)\left(\varepsilon_{r}-\varepsilon_{p}\right)}{\sigma_{p}\varepsilon_{p}}$$
where $\sigma_{p}$ and $\varepsilon_{p}$ are peak stress and peak strain, respectively. $\sigma_{r}$ and $\varepsilon_{r}$ are residual stress and residual strain, respectively. Taking the gel content as the x-axis and the confining pressure as the y-axis, the calculation results are plotted in the three-dimensional histogram (Figure 3). The graphic illustrates how the gel content and confining pressure have a significant impact on the brittleness index of CSG. The degree of brittleness rises with increased gel content and falls with higher confining pressure. The changing process from the upper left to the lower right in the figure represents the transformation process of the post-peak strain of CSG from hardening to softening.

### 2.3. Microscopic Characteristics

After curing and forming the CSG, the aggregate is wrapped in cement mortar and bonded together by the mortar. This cohesion plays the leading role when the cement mortar is not damaged. The friction resistance between aggregates progressively increased with the gradual failure of cementation (Figure 4a). The cohesion and friction are in equilibrium when loaded to the residual stage. Figure 4b demonstrates the CWFS theory proposed by V. Hajiabdolmajid [17,18]. This theory asserts that the change law of cohesion and friction angle with plastic strain can effectively reflect the strain hardening and softening characteristics. The CWFS model proved that the cohesion and friction angle remained unchanged in the elastic deformation stage. The cohesion weakens persistently, and the friction increases continuously, when loaded to the plastic stage. The friction resistance between particles will be fully developed when the cohesion decreases to 0.

### 2.4. Strength Parameters c and φ

The strain-hardening and strain-softening characteristics of geotechnical materials can be represented by the evolution law of the strength parameter that corresponds to the strain softening parameter in accordance with CWFS theory. Based on the Mohr–Coulomb strength criterion, cohesion c and friction angle *φ* were selected as the strength parameters. The general form of the Mohr–Coulomb strength criterion could be expressed as follows:(3)$$\sigma_{1}=\frac{1+\sin\varphi }{1-\sin\varphi }\sigma_{3}+\frac{2c\cos\varphi }{1-\sin\varphi }$$

The strain-softening parameters control the variation of the strength parameters of CSG. This study aims to reveal the variation law of cohesion and friction angle, so the selection of strain-softening parameters is very important. The ideal strain softening parameters should have the following characteristics. Firstly, during the process of strain hardening or softening, the strain softening parameters should change cumulatively and irreversibly. Secondly, the yield state of the material should be controlled by the coupling of the stress tensor and the strain softening parameters. This means that certain types of strain softening parameters correspond to only one strength criterion and do not change with the variation of the stress path. Finally, the strain softening parameters should contain multi-directional plastic deformation information to express the complete plastic state of the material. Under different confining pressures, the corresponding maximum principal strain should be different as the loading enters the plastic stage. Thus, the equivalent plastic strain is generally chosen as the strain softening parameter [48], the calculation expression is as follows:(4)$$\lambda =\sqrt{\frac{2}{3}\left(\varepsilon_{1}^{p}\varepsilon_{1}^{p}+\varepsilon_{2}^{p}\varepsilon_{2}^{p}+\varepsilon_{3}^{p}\varepsilon_{3}^{p}\right)}$$

The incremental form is as follows:(5)$$\lambda =\int \sqrt{\frac{2}{3}\left(d\varepsilon_{1}^{p}d\varepsilon_{1}^{p}+d\varepsilon_{2}^{p}d\varepsilon_{2}^{p}+d\varepsilon_{3}^{p}d\varepsilon_{3}^{p}\right)}$$

In a conventional tri-axial shear test, $\varepsilon_{2}^{p}=\varepsilon_{3}^{p}$, the Equation (4) can be written as follows:(6)$$\lambda =\sqrt{\frac{2}{3}\left(\varepsilon_{1}^{p}\varepsilon_{1}^{p}+2\varepsilon_{3}^{p}\varepsilon_{3}^{p}\right)}$$
where $\varepsilon_{1}^{p}$, $\varepsilon_{2}^{p}$, and $\varepsilon_{3}^{p}$ represent the plastic principal strain in three dimensions and the $d\varepsilon_{1}^{p}$, $d\varepsilon_{2}^{p}$, and $d\varepsilon_{3}^{p}$ represent the change rate of plastic principal strain in three dimensions. After the strain softening parameters are procured, the deviatoric stresses under different confining pressures are able to be acquired by the interpolation method. Specifically, the Mohr–Coulomb strength criterion is first simplified as follows:(7)$$\sigma_{1}=a\cdot \sigma_{3}+b$$

Through the conversion relationship with the Mohr–Coulomb criterion, cohesion *c* and friction angle *φ* are going to be acquired as follows:(8)$$\varphi =\arcsin\left(\frac{a-1}{a+1}\right)$$
(9)$$c=\frac{b\left(1-\sin\varphi \right)}{2\cos\varphi }$$

By plotting the calculated results in the 3D figure (Figure 5), it can be found that there exists a significant negative correlation between cohesion c and strain softening parameter λ, and a positive correlation between cohesion C and gel content. As for the friction angle *φ*, there is a distinctly positive correlation with the strain softening parameter λ and an inconspicuous correlation with the gel content. The above results are in reasonable agreement with Martin’s [14] research on rock materials. The difference is that at the initial damage point (the point where the strain softening parameter is 0), the initial friction angle obtained according to the strength and confining pressure of CSG is not 0. The cohesion will increase as the gel content rises, and the friction angle is influenced by the gel content and the particle breaking during the loading process. The correlation between the strength parameters and the strain softening parameters follows the CWFS strength criterion, and the difference is that the initial friction angle is not zero.

## 3. Particle Flow Analysis of Strength Parameter Evolution

### 3.1. Buildup of Model

V. Hajiabdolmajid [17,18] adopted a particle flow discrete element when creating their CWFS model. This model mainly elaborated on the development state of micro-cracks at different loading stages of rock, then revealed the internal relationship between the weakening of cohesion and the strengthening of friction. Additionally, Miao S J et al. [49,50] determined the key parameters for establishing the strength evolution law through particle flow simulation. Considering this, it is advisable to interpret the strength parameters defined at the microscopic level from the microscopic perspective. The structure of CSG material alters dramatically with the constant addition of gel content, which is produced when cement mortar bonds natural aggregates together (mostly aggregates without obvious edges and corners). Moreover, the structure of CSG is more complex than the rock materials. Thus, it is difficult to reflect the mechanical properties of the micro level of CSG by using BALL as the minimum element. In the model building section of this study, the CLUMP element is employed to simulate aggregate and the BALL element is employed to simulate mortar [51]. To reflect the impact of particle breakage on the macro mechanical properties of CSG, the CLUMP unit is converted into a flexible cluster that can represent crushing by using the FISH language programming function of PFC 5.0 commercial discrete element software (Figure 6).

Maintaining a similar aggregate structure, the CSG model with different gel contents was simulated by changing the porosity of the mortar and the bonding strength between particles. The models of CSG under different gel contents are shown in Figure 7.

By comparing the results of the mechanical test and simulation test, the final meso-mechanical parameters were determined, as shown in Table 3.

### 3.2. Acoustic Emission Simulation in Particle Flow

Under the action of external force, there exists the acoustic emission (AE) phenomenon because of the rapid release of local energy and the material’s emissions of transient elastic waves from the inside [52]. To implement AE monitoring during the particle flow loading process, Zhou Y et al. [53,54] and Cai et al. [55] compiled an AE moment tensor operation function into FISH language. According to the change of contact position and contact force, each component of the moment tensor is acquired by integrating the surrounding region of the micro-crack [56,57,58].

The specific equation of the operation is as follows:(10)$$M_{ij}=\sum_{S}\Delta F_{i}R_{j}$$
where surface *S* covers the micro-cracks generated by AE events. $\Delta F_{i}$ is the *i*th component of the contact force change, and $R_{j}$ is the *j*th component between the contact point and the AE event center. To save computing resources, the moment tensor with the largest scalar moment is selected to represent the AE event, which means it is unnecessary to calculate each time node within the range of AE continuous events. The specific calculation equation of scalar moment is as follows:(11)$$M_{0}={\left(\sum_{j=1}^{3}m_{j}^{2}/2\right)}^{2}$$
where $m_{j}$ is the eigenvalue of the moment tensor matrix. Furthermore, to speculate the failure intensity of the fracture event, the moment magnitude of the AE event should be calculated as follows:(12)$$M_{w}=\frac{2}{3}\log M_{0}-6$$

The moment magnitude of AE events can represent the severity of the fracture surface dislocation and the magnitude of energy released.

### 3.3. Particle Flow Simulation Results

Figure 8 shows the calculation results of the CSG particle flow model (the confining pressure is 600 kPa), and the curves attained by numerical calculation are in good agreement with the test curves. It should be mentioned that the strain hardening–softening transition characteristics in the post-peak stage are described, which proves the correctness of the numerical model. According to the energy tracer function of PFC5.0 software, Itasca Consulting Group Inc.: Minneapolis, MN, USA [59], the energy changes during the loading process of the model would be monitored. The energy created by relative slippage between particles is referred to as friction energy, whereas the energy created when a particle’s bond dissolves is referred to as bonding energy. From the perspective of energy distribution, before the stress reaches the peak, a significant quantity of bonding energy is released, which is significantly more than friction energy. This phenomenon indicates that the failure of cement-based materials before the peak is what primarily causes this instability. After the peak, with the failure and instability of cement-based materials, the friction energy is constantly rising due to the friction resistance between particles. The change in the energy curve also confirms the mutual transformational relationship between cohesion and friction. 

According to the AE monitoring results, there are rarely AE events in the initial elastic stage. After entering the damage stage, the AE frequency and intensity increase continuously, reaching the maximum value at the peak. The post-peak AE intensity decreases with a reduction in the stress to the residual stress and maintains stability. The frequency and intensity are maintained at a low level and tend to be stable, even if acoustic emission events continue to occur. Five characteristic intensities were defined according to the development law of AE events (Figure 8). Points A, B, C, D, and E represent the initial crack onset intensity, the plastic development intensity, the peak intensity, the peak drop intensity, and the residual intensity, respectively.

### 3.4. AE Spatio-Temporal Distribution

The AE spatio-temporal distribution is shown in Figure 9. When the gel content is at a low level, the cementing material between aggregates breaks rapidly, and most AE events also gather in this location. As the aggregate lost the protection of cementation, there occurred a large number of particle breakages, which made the friction between the aggregates continuously increase, resulting in an inconspicuous downward trend in the post-peak stage. When the gel content is at a high level, the cement-based material is not completely destroyed in the post-peak stage, and the particle breakage phenomenon rarely occurs. As the cement-based material is further damaged, the post-peak curve drops with significant residual strength. The above results also illustrate the characteristics of strain hardening–softening transition of CSG from a mesoscopic perspective. The cement-based material between the aggregates is damaged first, which leads to a rapid decrease in cohesion. After the bond between the particles is broken, the friction between the particles resists the external force, leading to an increase in the friction angle. Furthermore, the crushed aggregate remains stable through friction resistance, which further increases the friction angle. The evolution of cohesion and friction angle also reaches a steady state when the post-peak bond failure’s development ends.

## 4. Evolution Law of Strength Parameters

### 4.1. Evolution Model

Although many valuable improvements in the theory and application of the functional relationship between strength parameters and strain softening parameters have been achieved, most of the evolutionary models’ establishment has been based on exponential functions and has varied according to different research objects. According to the mechanical properties of CSG and the variation characteristics of strength parameters, the evolution law between cohesion and strain softening parameters can be expressed by the following equation:(13)$$c=c_{i}-\left(c_{i}-c_{r}\right)\cdot \left(1-e^{-{\left(\lambda /\lambda_{c}^{p}\right)}^{2}}\right)$$

The evolution law of friction angle can be expressed as follows:(14)$$\varphi =\varphi_{i}+\left(\varphi_{r}-\varphi_{i}\right)\cdot \left(1-e^{-\lambda /\lambda_{\varphi }^{p}}\right)$$

In Equations (13) and (14), each parameter has a clear physical meaning. $c_{i}$, $c_{r}$, $\varphi_{i}$, $\varphi_{r}$, $\lambda_{c}^{p}$, and $\lambda_{\varphi }^{p}$ represent the initial cohesion, residual cohesion, initial friction angle, residual friction angle, strain-softening parameters under cohesion, and internal friction angle stability, respectively. According to the research method of Miao S J et al. [49], the critical strain-softening parameters $\lambda_{c}^{p}$ and $\lambda_{\varphi }^{p}$ have been calculated and solved from the mesoscopic perspective.

According to the previous analysis, it is practicable to reveal the fracture evolution process of CSG from the mesoscopic perspective. Figure 10 shows the generation mechanisms of tension cracks and shear cracks between particles. The particles produce shear cracks under the action of friction and tensile cracks under the action of tension. Consequently, the number of tensile cracks and shear cracks can represent the change law of cohesion and friction force, and they can also be considered the product of cohesion and friction, respectively. The change in the number of tension cracks and shear cracks corresponds to the variation of cohesion and friction angle. Based on the corresponding growth law of cracks stabilization, it is possible to compute the plastic strains $\lambda_{c}^{p}$ and $\lambda_{\varphi }^{p}$.

Figure 11 shows the growth law of tensile cracks and shear cracks in the discrete element model of CSG under different gel contents. The axial plastic principal strain, which corresponds to the change in *c* and *φ*, is the strain that causes the fracture curve to expand until it reaches a stable inflection point. The radical plastic principal strains $\varepsilon_{3c}^{p}$ and $\varepsilon_{3\varphi }^{p}$ were then obtained from the axial–radical strain curves. Therefore, according to Equation (4), $\lambda_{c}^{p}$ of CSG under 20 kg/m^3^, 60 kg/m^3^, and 100 kg/m^3^ is 0.133, 0.083, and 0.038, and $\lambda_{\varphi }^{p}$ is 0.153, 0.094, and 0.044, respectively.

### 4.2. Model Validation

We took the CSG with gel contents of 20 kg/m^3^, 60 kg/m^3^, and 100 kg/m^3^ as examples to verify the established strength parameter evolution model. The parameters involved in the model are shown in Table 4.

As shown in Figure 12, Equations (13) and (14) can well describe the nonlinear evolution law of cohesion and friction angle. As reported in Table 4 and Figure 12, the degradation rate of early cohesion is accelerated, and the residual cohesion is increased with the increase of gel content. The increase of gel content also ascends the growth rate of friction angle, while there is no evident correlation between gel content and residual friction angle. Overall, the strength parameter evolution model established in this paper accurately reflects the changing process of cohesion and internal friction angle with the strain softening parameters. Moreover, it explains the microscopic failure mechanism of cemented sand and gravel with different cementation contents.

## 5. Statistical Damage Model of CSG

### 5.1. Damage Model Construction

It is crucial to generate the proper micro-strength function when building a statistical damage constitutive model. Previous studies have proved that taking classical strength theory as the micro-strength function is a valid research method. However, it is difficult for the classical strength theory to characterize the damage evolution law of materials effectively. Consequently, the strength theory can be modified according to the damage and fracture mechanism of materials, after which the damage statistical constitutive model of CSG can be built.

The Lemaitre strain equivalence hypothesis is as follows [60]:(15)$$\bar{\sigma_{ij}}=\frac{\sigma_{ij}}{\left(1-D\right)}$$
where $\sigma_{ij}$ represents the nominal stress measured during the test and $\bar{\sigma_{ij}}$ represents the effective stress supported by the portion of the material that is undamaged.

After taking residual stress into account, Equation (15) can be established as follows:(16)$$\sigma_{1}=\bar{\sigma_{1}}\left(1-D\right)+\sigma_{r}D$$

The stress–strain relationship of the undamaged part obeys the generalized Hooke’s law:(17)$$\bar{\sigma_{1}}=E\bar{\varepsilon_{1}}+\mu (\bar{\sigma_{2}}+\bar{\sigma_{3}})$$

The residual stress no longer satisfies generalized Hooke’s law, but still applies the Mohr–Coulomb strength criterion. Hence the residual stress can be expressed as follows: (18)$$\sigma_{r}=\frac{\left(1+\sin\varphi_{r}\right)\sigma_{3}+2c_{r}\cos\varphi_{r}}{1+\sin\varphi_{r}}$$
where $c_{r}$ and $\varphi_{r}$ are residual cohesion and residual friction angle, respectively.

According to the principle of deformation coordination, the nominal axial strain $\varepsilon_{1}$ of the material, the microscopic axial strain $\bar{\varepsilon_{1}}$ of the undamaged part, and the microscopic axial strain $\varepsilon_{1r}$ of the damaged part are equal to each other:(19)$$\varepsilon_{1}=\bar{\varepsilon_{1}}=\varepsilon_{1r}$$

Assuming that there is no lateral damage, that is, $\sigma_{2}=\bar{\sigma_{2}}$ and $\sigma_{3}=\bar{\sigma_{3}}$, the *σ*_1_ can be obtained by combining Equations (16)–(19) as follows:(20)$$\sigma_{1}=E\varepsilon_{1}\left(1-D\right)+\omega D+\mu \left(\sigma_{2}+\sigma_{3}\right)$$

Among these, the *ω* can be described as follows:(21)$$\omega =\frac{\left(1+\sin\varphi_{r}\right)\sigma_{3}+2c_{r}\cos\varphi_{r}}{1-\sin\varphi_{r}}-\mu \left(\sigma_{2}+\sigma_{3}\right)$$

According to the idea put forward by Omid Pourhosseini et al. [20], before the material enters the damage stage, cohesion resists all external pressures. The constant loss of cohesion causes the material to deform and become unstable, which causes the material to become damaged. Therefore, the reduction of cohesion is closely related to the damage of the material. The strength reduction after damage only depends on the decrease of cohesion and has no direct relationship with the change of friction angle. The Mohr–Coulomb strength criterion should be chosen as the micro-element strength. Cohesion remains unchanged in the elastic stage and continues to evolve in the plastic stage. Consequently, the micro-element strength function is set as follows [39,44]:(22)$$F=\{\begin{array}{cc} \bar{\sigma_{1}}\left(1-\sin\varphi \right)-\bar{\sigma_{3}}\left(1+\sin\varphi \right)-2c\cos\varphi & \varepsilon_{1}\leq \varepsilon_{1c}^{p}\\ \bar{\sigma_{1}}\left(1-\sin\varphi \right)-\bar{\sigma_{3}}\left(1+\sin\varphi \right)-2c\left(\lambda \right)\cos\varphi & \varepsilon_{1}>\varepsilon_{1c}^{p} \end{array}$$

The probability density based on the Weibull distribution function is as follows:(23)$$P(F)=\frac{m}{F_{0}}{\left(\frac{F}{F_{0}}\right)}^{m-1}\cdot e^{-{\left(F/F_{0}\right)}^{m}}$$

Let α=1−sinφ, β=1+sinφ and $\gamma =\{\begin{array}{cc} 2c\cos\varphi & \varepsilon_{1}\leq \varepsilon_{1c}^{p}\\ 2c\left(\lambda \right)\cos\varphi & \varepsilon_{1}>\varepsilon_{1c}^{p} \end{array}$, then:(24)$$F=\alpha \bar{\sigma_{1}}-\beta \bar{\sigma_{3}}-\gamma$$

Substituting Equation (17) into Equation (24), let $\sigma_{2}=\bar{\sigma_{2}}$ and $\sigma_{3}=\bar{\sigma_{3}}$, the equation can be obtained as follows:(25)$$F=\alpha \left[E\varepsilon_{1}+2\mu \left(\sigma_{2}+\sigma_{3}\right)\right]-\beta \sigma_{3}-\gamma$$

If the influence of the damage threshold is considered, the damage variable in the statistical damage evolution model can be expressed as follows:(26)$$D=\{\begin{array}{cc} 1-\exp\left[-{\left(\frac{F}{F_{0}}\right)}^{m}\right] & F\geq 0\\ 0 & F<0 \end{array}$$

### 5.2. Weibull Parameter-Solving Process

In the traditional statistical damage constitutive model, Weibull distribution parameters can be obtained directly by solving the integral method of generalized Hooke’s law. Nevertheless, considering the residual stress, the damage constitutive model is more complicated. Thus, this method is no longer applicable and can be solved by the extreme points in the stress–strain curve. In Equation (21), the partial derivative $\varepsilon_{1}=\varepsilon_{m}$ can be obtained as follows:(27)$$\frac{\partial \sigma_{1}}{\partial \varepsilon_{1}}\left(\varepsilon_{1}=\varepsilon_{m}\right)=E\left(1-D_{p}\right)+\left(\omega -E\varepsilon_{m}\right)\frac{\partial D}{\partial \varepsilon_{1}}=0$$
where $\varepsilon_{m}$ is the axial peak strain. To simplify the calculation, *γ* is assumed to be a constant, then:(28)$$\frac{\partial D}{\partial \varepsilon_{1}}=\frac{\partial D}{\partial F}\cdot \frac{\partial F}{\partial \varepsilon_{1}}=m{\left(\frac{F}{F_{0}}\right)}^{m-1}\cdot \exp\left[-{\left(\frac{F}{F_{0}}\right)}^{m}\right]\cdot \frac{\alpha E}{F_{0}}$$

Substitute Equation (26) into Equation (28), and after sorting, the equation can be obtained as follows:(29)$$\frac{\partial D}{\partial \varepsilon_{1}}=\left(1-D\right)\cdot \frac{m}{F_{0}}\ln\left(1-D\right)\cdot \alpha E$$

Combined with Equations (27) and (29), the equation can be obtained as follows:(30)$$m=\frac{F_{m}}{\alpha \left(E\varepsilon_{m}-\omega \right)\ln\left(1-D_{m}\right)}$$
where $F_{m}$ represents the micro-element strength when the stress reaches its peak, which can be obtained according to Equation (22):(31)$$F_{m}=\left(1-\sin\varphi \right)\sigma_{m}-\left(1+\sin\varphi \right)\sigma_{3}-2c\left(\lambda_{m}\right)\cos\varphi$$
where $\sigma_{m}$ is the peak axial stress, and $\lambda_{m}$ is the strain softening parameter when the strain reaches the peak.

$D_{m}$ represents the damage variable when the stress reaches its peak, which can be calculated according to Equation (20) as follows:(32)$$D_{m}=\frac{\sigma_{m}-\mu \left(\sigma_{2}+\sigma_{3}\right)-E\varepsilon_{m}}{\omega -E\varepsilon_{m}}$$

If (26), (30), (31), (32) are combined, *F*_0_ can be acquired as follows:(33)$$F_{0}=F_{m}{\left[-\ln\left(1-D_{m}\right)\right]}^{-\frac{1}{m}}$$

### 5.3. Weibull Parameter-Solving Process

As shown in Figure 13, both the m and *F*_0_ can impact the variation rule of the stress–strain curve, especially in the post-peak stage. The peak strength and the post-peak stress drop rate will increase with the growth of m, reflecting the strain hardening–softening transformation characteristics. The rise of *F*_0_ will increase the Young’s modulus and peak stress of the curve. It can be seen that the stress–strain curve can be changed by controlling parameters m and *F*_0_, especially in the post-peak stage. This phenomenon reflects the influence of gel content on the mechanical properties of CSG.

As shown in Figure 14, both parameters m and F_0_ affect the change of damage variable D. The rise of m makes the damage evolution process transform from two stages to three stages and significantly increases the growth rate of the intermediate stage. *F*_0_ also affects the growth rate of damage variables. The larger *F*_0_ is, the slower the growth rate of damage variables will be.

To deliberate the applicability of the constitutive model established in this study, triaxial experimental data of CSG with gel contents of 20 kg/m^3^ (without significant residual strength) and 100 kg/m^3^ (with significant residual strength) were selected. Additionally, the corresponding Weibull distribution parameters were solved to construct the damage constitutive model and the Weibull distribution is listed in Table 5. The test curve is consistent with the theoretical calculation curve and has merit applicability to the CSG with different gel contents (Figure 15). When the gel content is 20 kg/m^3^, the model is in basically good agreement with the experimental data, and the post-peak strain hardening degree increases with the increase of confining pressure. When the gel content is 40 kg/m^3^ and the confining pressure is 300 kPa, the post-peak curve has an obvious drop phenomenon, and with the increase of confining pressure, the post-peak curve becomes stable. When the gel content is 100 kg/m^3^, the model curve fits well with the test results, and the drop degree of the model post-peak curve is basically the same as the test data. The constitutive model is particularly adept at illustrating the total stress–strain curve and the strain hardening–softening transition properties of the CSG with various gel compositions, reflecting the microscopic failure mechanism of CSG. Therefore, the damage model based on the strength evolution law fully reflects the strain-hardening and strain-softening characteristics of CSG and reveals the relationship between cohesion weakening and material damage.

## 6. Conclusions

Based on the complex mechanical behavior characteristics of CSG, this study expounded on the microscopic failure mechanism of CSG. Through the established particle discrete element model, the evolution mechanism of cohesion and friction angle of CSG was revealed. Consequently, the evolution law of strength parameters was explored, and a statistical damage constitutive model of CSG was established according to the evolution process of cohesion. The main conclusions of this study are as follows:

(1) With the assistance of the CWFS theory, we can see that the essential phenomena of strain-softening and strain-hardening are equivalent to the evolution law of strength parameters. The cohesion decreases with the increment of the strain-softening parameter and increases with the increment of the gel content. The friction angle increases with the increment of the strain-softening parameter and has an inapparent correlation with the gel content.

(2) The cement-based material was the first to break and release a large amount of bonding energy, which led to the reduction of cohesion. The growth of friction resistance between particles and the friction action after aggregate breakage makes the friction angle increase continuously. Thus, the released friction energy keeps increasing, and the bonding energy gradually tends toward stability. According to the AE distribution law, the AE’s intensity and frequency increase before the peak, decrease after the peak and reach a steady state at the residual stage. AE events first occur in cement-based materials, where, due to the loss of the protective effect, aggregate particle breakage can occur. With the increase in gel content, the aggregate is not easily broken, and the failure mode is close to brittle failure.

(3) The evolution model of cohesion and friction angle is established, and each parameter in the model possesses a corresponding physical meaning. The critical strain-softening parameters of cohesion and friction angle are determined in conformity with the curve of tension crack and shear crack in PFC.

(4) Taking the Mohr–Coulomb strength criterion and considering cohesion loss as the micro strength distribution function, a damage statistical constitutive model of CSG was constructed. The model can fully picture the strain-hardening and strain-softening characteristics of CSG and can indicate the internal relationship between cohesion weakening and material damage. It has been demonstrated in this paper that the statistical damage constitutive model established through this study offers excellent applicability and accuracy.

## Figures and Tables

**Figure 1 materials-16-00542-f001:**
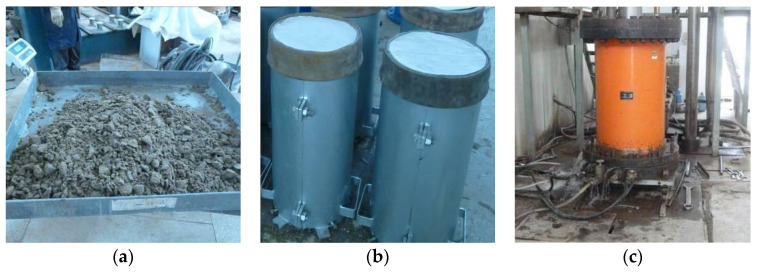
Tri-axial test process of CSG: (**a**) material mixing; (**b**) filling in the model; (**c**) loading test.

**Figure 2 materials-16-00542-f002:**
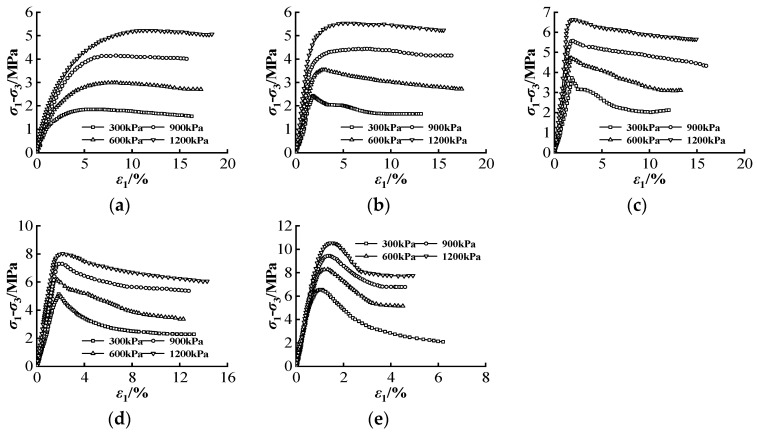
The tri-axial strain–stress curve of CSG under different gel contents: (**a**) 20 kg/m^3^; (**b**) 40 kg/m^3^; (**c**) 60 kg/m^3^ (**d**) 80 kg/m^3^; and (**e**) 100 kg/m^3^.

**Figure 3 materials-16-00542-f003:**
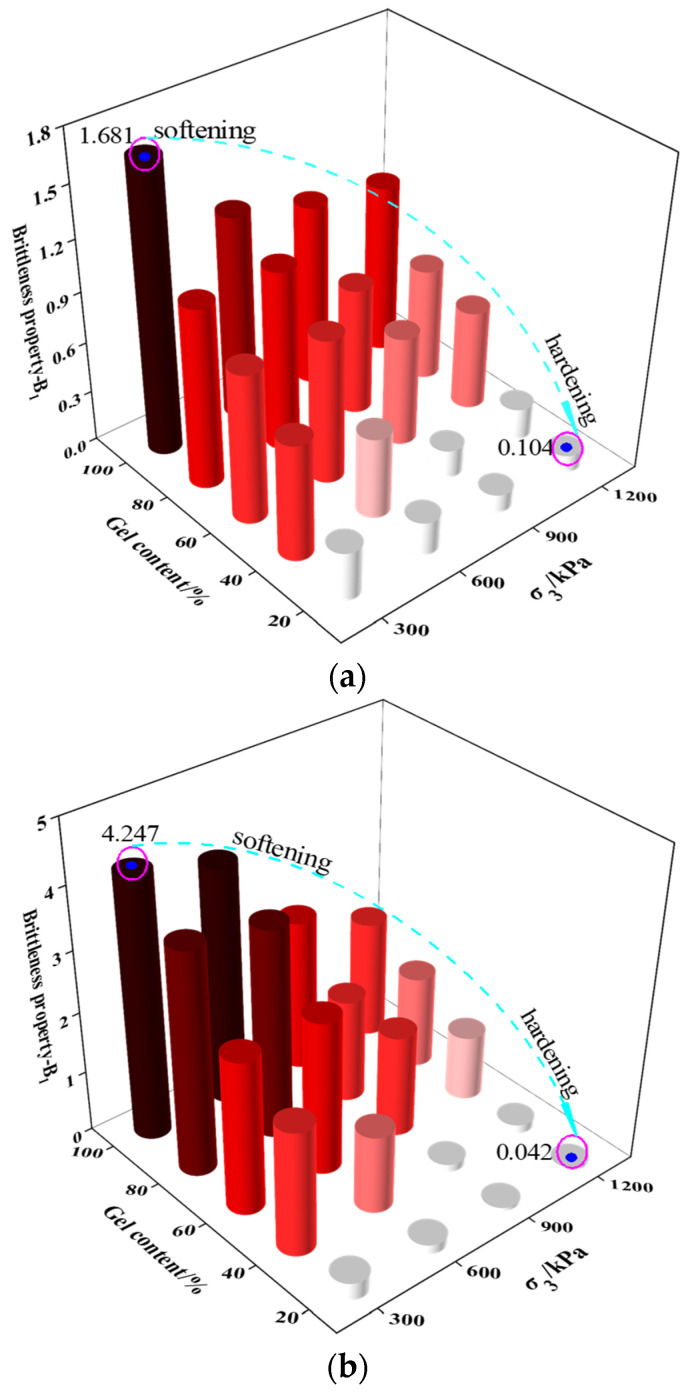
Brittleness index of CSG: (**a**) *B*_1_; (**b**) *B*_2_.

**Figure 4 materials-16-00542-f004:**
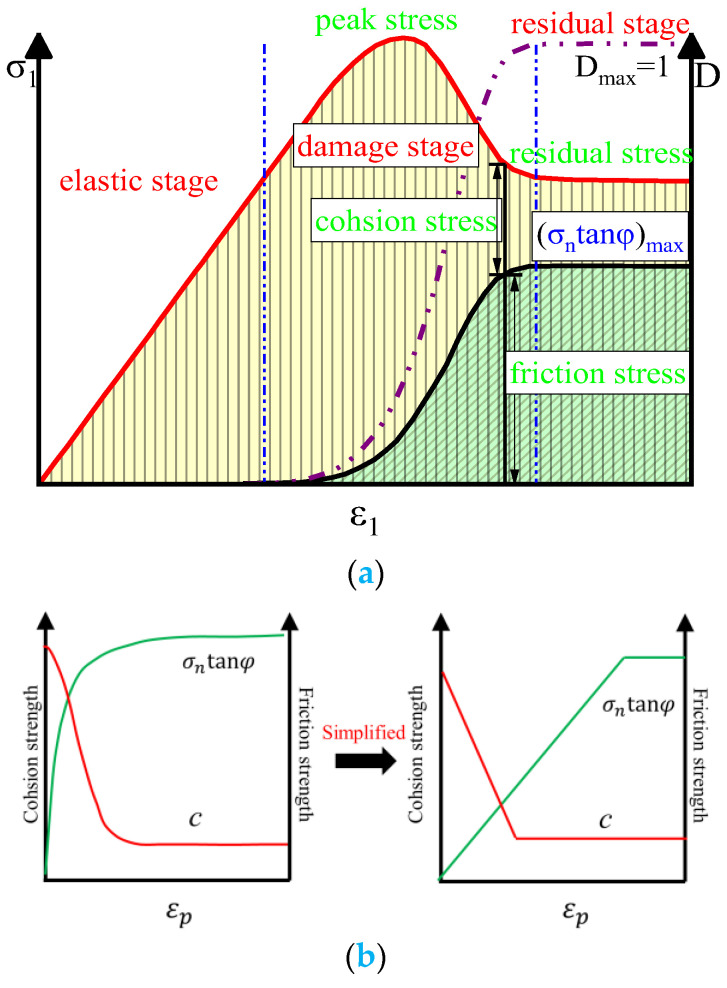
Related theory of strength parameter evolution: (**a**) Evolution law of cohesion and friction stress; (**b**) CWFS theory curve.

**Figure 5 materials-16-00542-f005:**
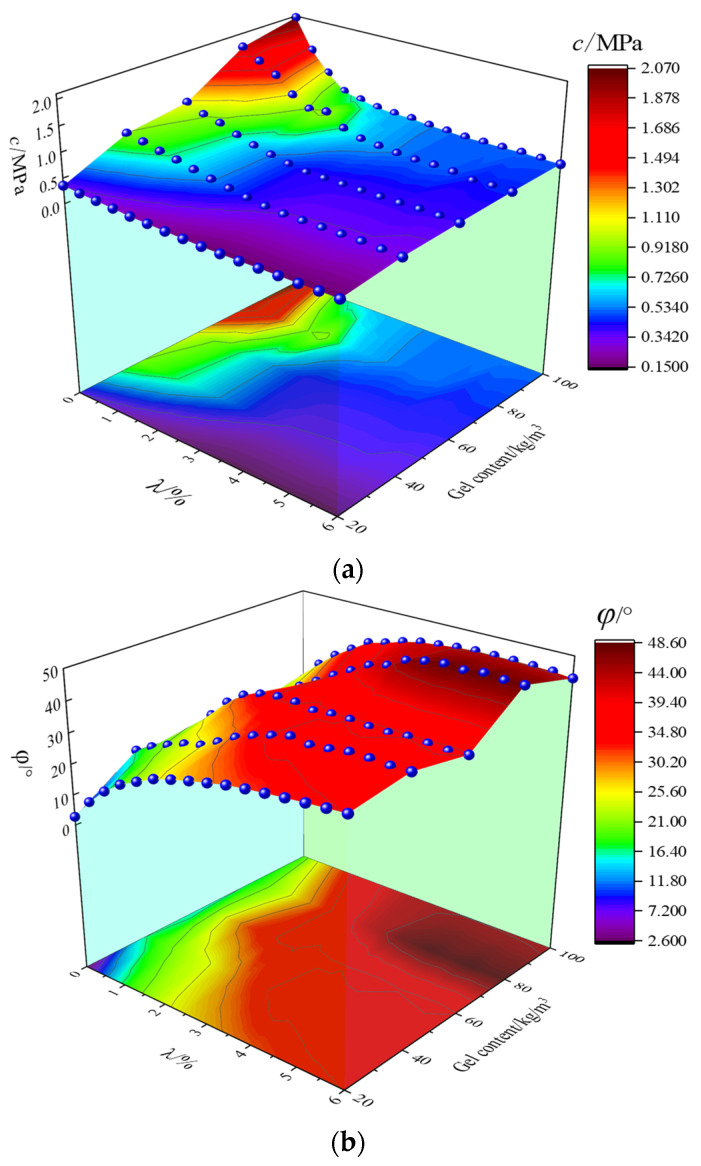
Three-dimensional surface diagram of strength parameters: (**a**) cohesion *c*; (**b**) friction angle *φ*.

**Figure 6 materials-16-00542-f006:**
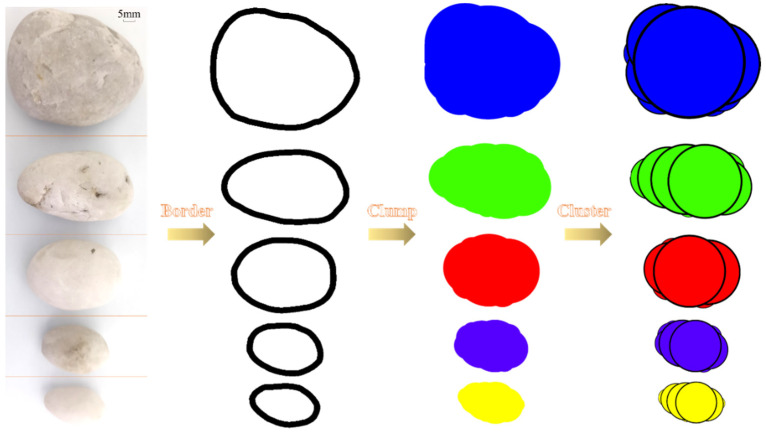
Discrete element modeling of aggregates with different particle sizes.

**Figure 7 materials-16-00542-f007:**
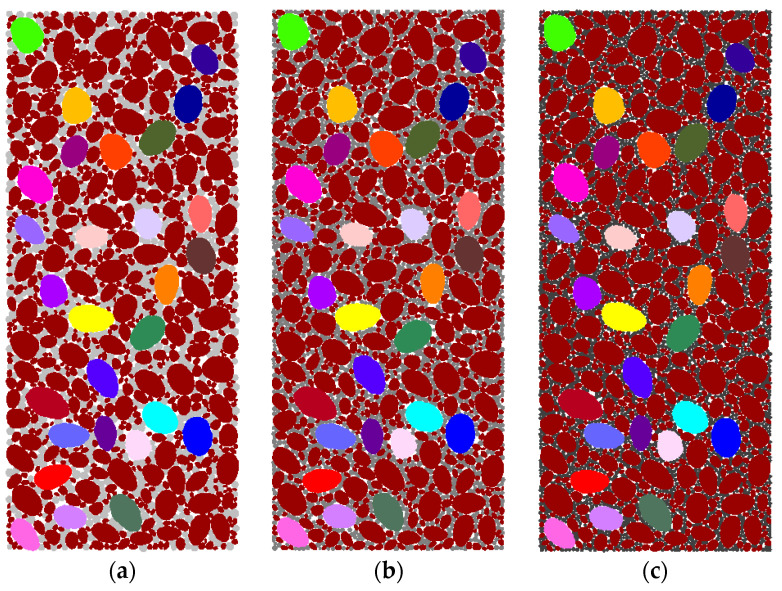
Particle flow model of CSG: (**a**) 20 kg/m^3^; (**b**) 60 kg/m^3^; (**c**) 100 kg/m^3^.

**Figure 8 materials-16-00542-f008:**
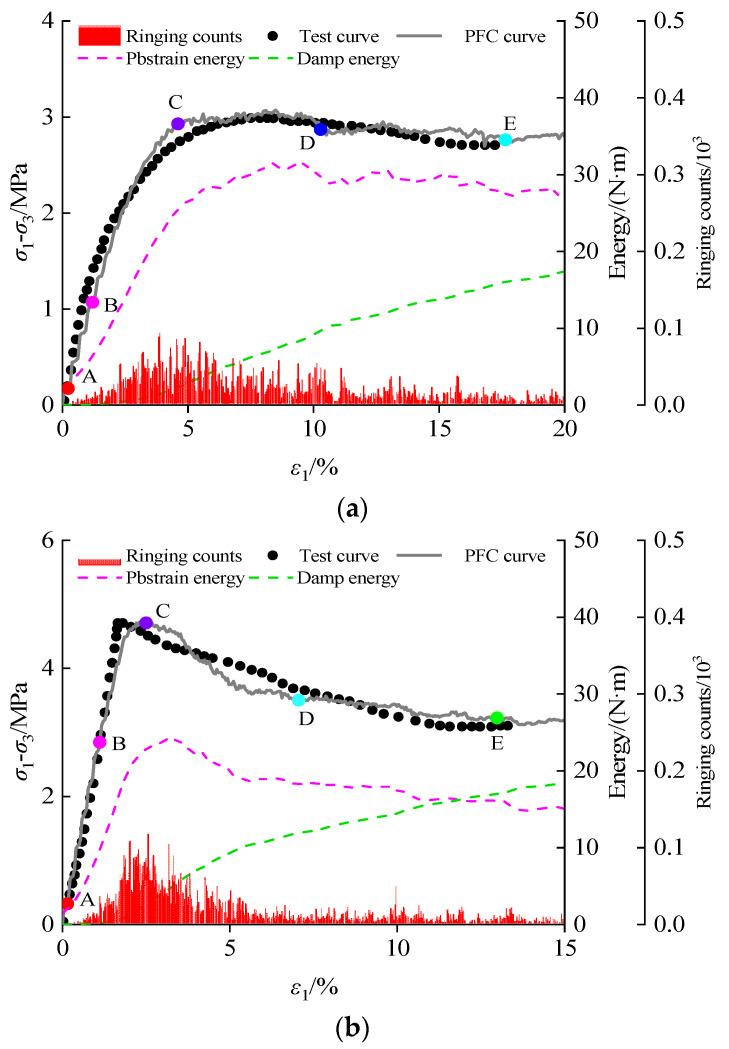
Test results and particle flow simulation results of CSG: (**a**) 20 kg/m^3^; (**b**) 60 kg/m^3^; (**c**) 100 kg/m^3^.

**Figure 9 materials-16-00542-f009:**
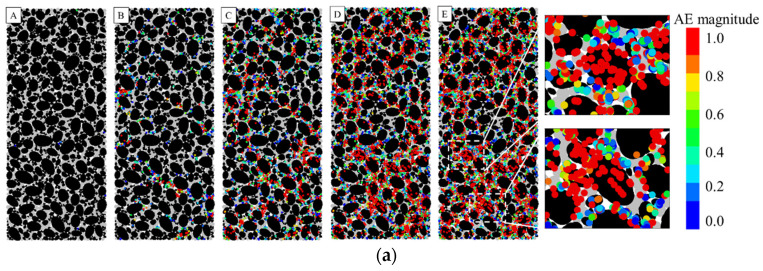
The spatio-temporal distribution of AE (Acoustic emission) signal of CSG with different gel contents: (**a**) 20 kg/m^3^; (**b**) 60 kg/m^3^; (**c**) 100 kg/m^3^.

**Figure 10 materials-16-00542-f010:**
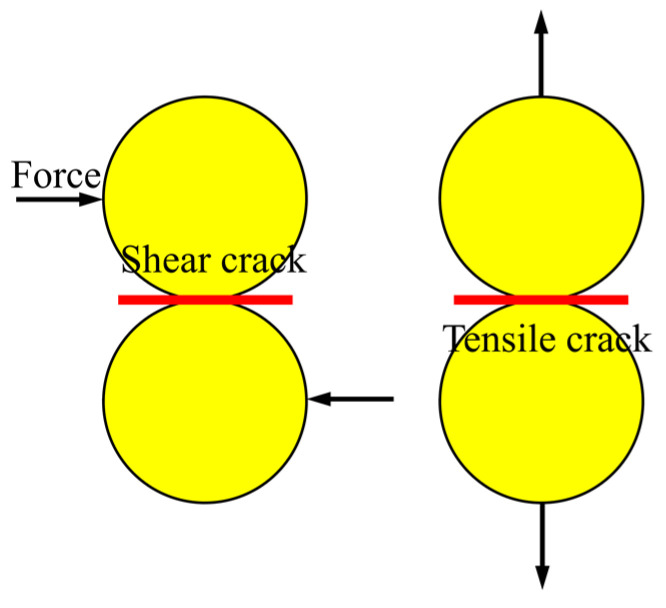
Mechanism of micro-crack generation in discrete element model.

**Figure 11 materials-16-00542-f011:**
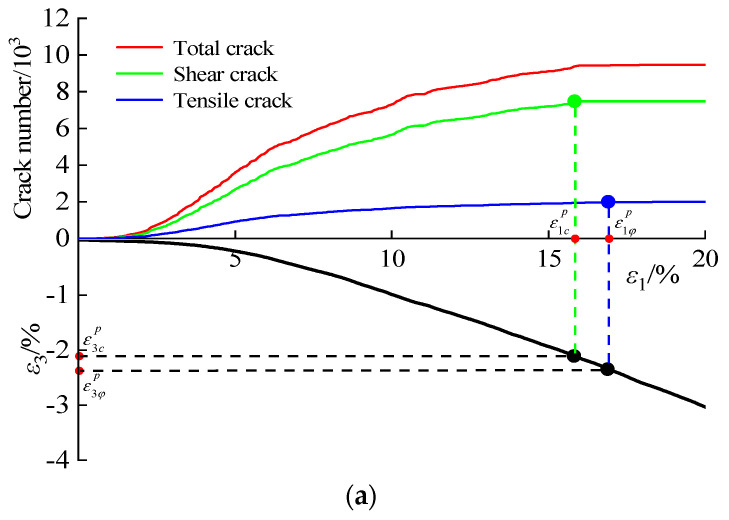
The law of micro-crack development: (**a**) 20 kg/m^3^; (**b**) 60 kg/m^3^; (**c**) 100 kg/m^3^.

**Figure 12 materials-16-00542-f012:**
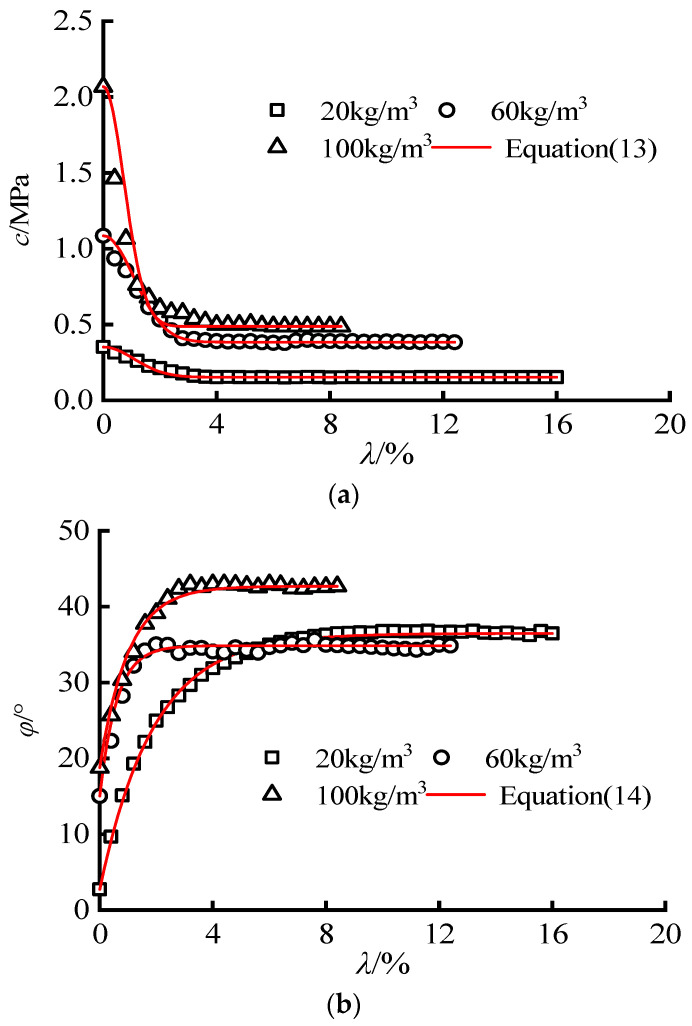
Strength parameter evolution equation of CSG: (**a**) cohesion c; (**b**) friction angle *φ*.

**Figure 13 materials-16-00542-f013:**
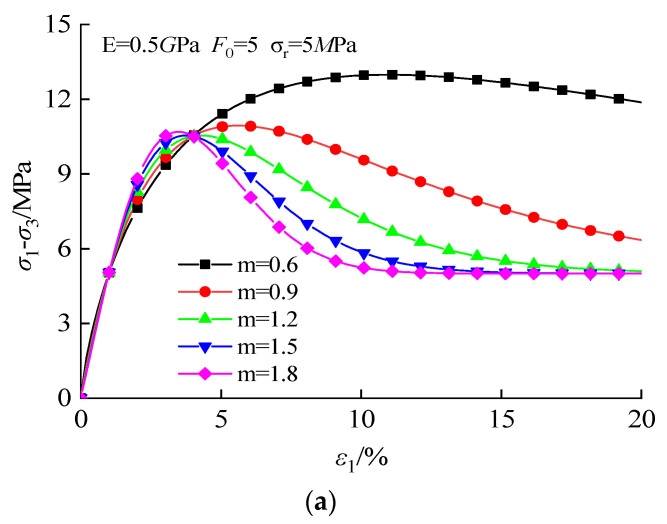
Effect of the Weibull parameter on the stress–strain curve: (**a**) m; (**b**) F_0_.

**Figure 14 materials-16-00542-f014:**
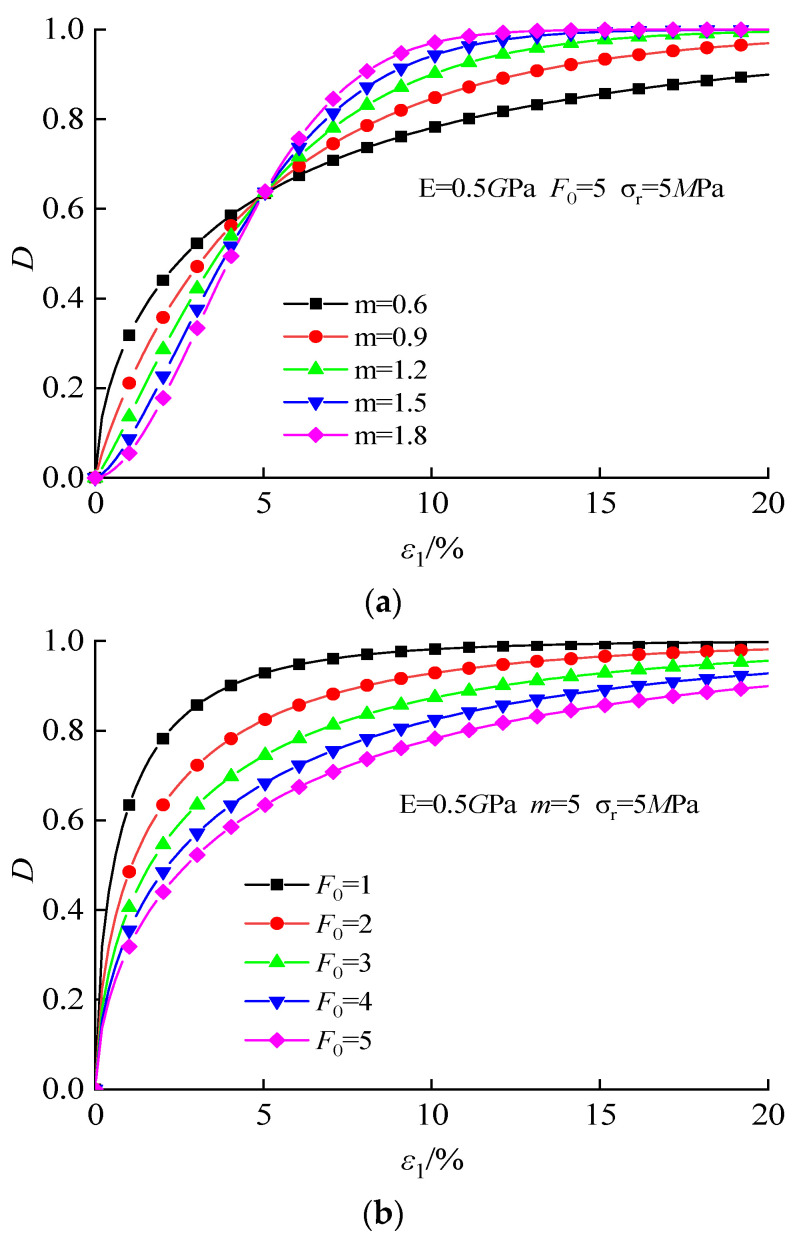
The influence of the Weibull parameter on damage variable D: (**a**) m; (**b**) F_0_.

**Figure 15 materials-16-00542-f015:**
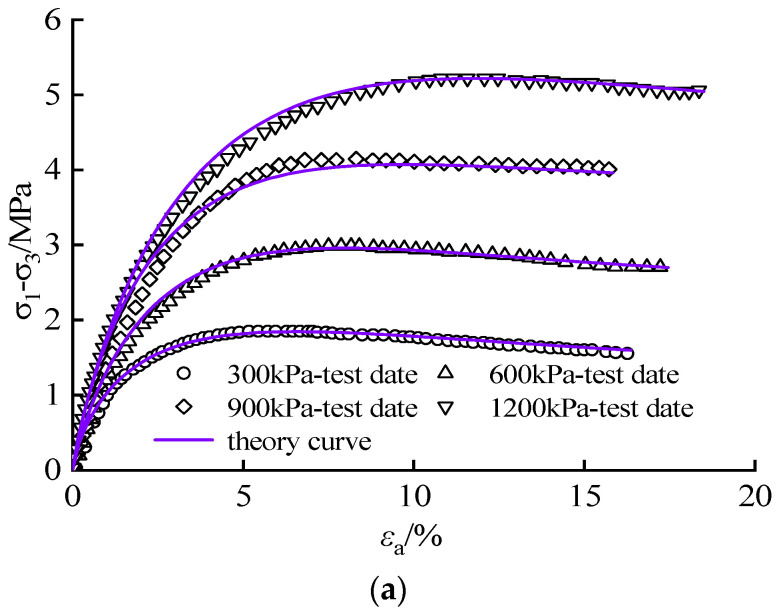
Comparison between the test curve and theory curve: (**a**) 20 kg/m^3^; (**b**) 60 kg/m^3^; (**c**) 100 kg/m^3^.

**Table 1 materials-16-00542-t001:** Aggregate gradation.

Grading/mm	<1	1~5	5~10	10~20	20~40
Proportion/%	20	10	20	20	30

**Table 2 materials-16-00542-t002:** Mixture proportions of CSG (kg/m^3^).

Gel Content	Ratio	Water	Cement	Fly Ash	Aggregate
20	1.0	20	10	10	2130
40	1.0	40	20	20	2130
60	1.0	60	30	30	2130
80	1.0	80	40	40	2130
100	1.0	100	50	50	2130

**Table 3 materials-16-00542-t003:** Mesoscopic parameters of particle flow models.

Materials	Gel Content/kg/m^3^	*E*/GPa	*k*_n_/GPa	*k*_s_/GPa	*μ*	Porosity/%
Mortar	20	69	0.75	0.69	0.4	0.01
	60	76	0.85	0.77	0.4	0.015
	100	81	0.99	0.89	0.4	0.02
Aggregate	-	62.5	1.84	1.67	0.5	-

**Table 4 materials-16-00542-t004:** Strength evolution parameters under different gel content.

Gel Content/kg/m^3^	20	60	100
*c*_i_/MPa	352.32	1523.96	2068.32
*φ*_i_/°	5.74	15.05	18.80
*c*_r_/MPa	83.03	540.66	900.15
*φ*_r_/°	36.83	34.45	43.68
$\lambda_{c}^{p}$/%	13.26	8.25	3.84
$\lambda_{\varphi }^{p}$/%	15.31	9.37	4.35

**Table 5 materials-16-00542-t005:** The Weibull distribution parameters.

Gel Content/kg/m^3^	σ_3_/kPa	*m*	*F* _0_	*E*/GPa	*σ_r_*/MPa
20	300	0.653	229.25	0.085	1.251
	600	0.813	332.34	0.088	2.457
	900	0.823	303.41	0.091	3.707
	1200	0.847	482.58	0.095	4.431
60	300	0.541	85.57	0.195	1.653
	600	0.553	108.21	0.201	2.723
	900	0.907	202.35	0.205	3.854
	1200	1.406	349.36	0.217	5.231
100	300	1.521	125.08	1.057	2.298
	600	1.619	127.45	1.058	5.152
	900	1.623	129.33	1.118	6.752
	1200	1.741	135.24	1.138	7.731

## Data Availability

The data presented in this study are available on request from the corresponding author.

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
