# Peer review of "A Study of Strength Parameter Evolution and a Statistical Damage Constitutive Model of Cemented Sand and Gravel"

_materials, 2023, doi:10.3390/ma16020542_

Round 1
Reviewer 1 Report
Very well articulated manuscript and can make a marked contribution to the construction industry. A few minor language corrections are suggested. The reviewer would also like to suggest the authors to revise the title of the manuscript. Good work!

Author Response
Comment: Very well articulated manuscript and can make a marked contribution to the construction industry.
Response: The authors appreciate the respected Reviewer #1 for his/her insightful review of this manuscript and positive perspective. The authors have provided detailed responses to your merit comments as follows:
Point 1: A few minor language corrections are suggested.
Response 1: Thank you for your suggestions on the language of this article, which are of great help to improve the quality of our manuscript. Following the respected reviewer’s comment, the authors have made modifications one by one and the modifications have been marked in red. Additionally, the authors have invited a foreign expert to contribute to the manuscripts' technical and grammatical revision. Several reviews were given to the manuscripts, and written English was modified. Hence, the authors modified all the manuscript text, including the spelling of the words and common grammar errors that the respected reviewers have mentioned.
Point 2: The reviewer would also like to suggest the authors to revise the title of the manuscript.
Response 2: The authors appreciate the respected reviewer’s comment. The title of the manuscript is modified.
The previous title: “A Research on Strength Parameter Evolution and Damage Characteristics of Cemented Sand and Gravel”.
The modified title: “A Research on Strength Parameter Evolution and Statistical Damage Constitutive Model of Cemented Sand and Gravel”.
I have revised the suggestions you put forward in the manuscript one by one, and marked each modification position.
Reviewer 2 Report
The article is well written and the quality of article can be improved significantly by addressing the following concerns:
1. Literature section can be improved by citing the latest documents i.e. 2022
2. The figures and tables need explanation in the text for better understanding
3. How the authors have decided the mixture proportion? any literature proof.
4. High quality of figures to be used. For ex. figure 3, is difficult to read.
Author Response
Comment: The article is well written and the quality of article can be improved significantly by addressing the following concerns.
Response: The authors want to express their appreciation to the respected Reviewer #2 for her/his positive comments and insightful questions that helped us to improve this manuscript substantially. The authors have provided detailed answers and actions to the reviewer’s comments as follows:
Point 1: Literature section can be improved by citing the latest documents i.e. 2022.
Response 1: Thanks for your merit suggestions. This study cited some classical literature and some of them have been replaced with the latest literature carefully. Still, some other classical literature is not easy to be replaced to better support the core points of this manuscript. The replaced literature has been marked in red.
Point 2: The figures and tables need explanation in the text for better understanding.
Response 2: The authors agree with the respected reviewer’s comment and appreciate the suggestion. All the pictures and tables have been explained in detail, and their corresponding positions in the text have been carefully checked. The changes have been marked in red.
Point 3: How the authors have decided the mixture proportion? any literature proof.
Response 3: Thanks for your comments. The selection of gel content is mainly based on the SL678-2014 Technical guideline for cemented granular material dams. In order to reflect the influence of gel content on mechanical properties of cemented sand and gravel, five groups of gel content of 20kg/m3, 40kg/m3, 60kg/m3, 80kg/m3, and 100kg/m3 were selected by equidifference, and the samples were prepared according to the mixture proportion in the technical guideline. Some related references are as follows:
[1] Yang, J. Cai, X. Guo, X. Zhao, J. Effect of Cement Content on the Deformation Properties of Cemented Sand and Gravel Ma-terial, Appl. Sci-Basel. 2019, 6, 1–16. https://sci-hub.se/10.3390/app9112369
[2] Wu, M. Du, B. Yao, Y. He, X. An experimental study on stress-strain behavior and constitutive model of hard fill material, Sci. China. Phys. Mech. 2011, 54, 2015–2024.https://sci-hub.se/10.1007/s11433-011-4518-3
[3] Lohani, T.N. Kongsukprasert, L. Watanabe, K. Tatsuoka, F. Strength and deformation properties of compacted ce-ment-mixed gravel evaluated by triaxial compression tests, Soils Found. 2004, 44, 95–108. https://sci-hub.se/10.3208/sandf.44.5_95
Point 4: High quality of figures to be used. For ex. figure 3, is difficult to read.
Response 4: Sorry for the confusion. The authors have redrawn Figure 3 by removing the numbers that represent the values and only keeping the maximum and minimum values. The higher the gel content, the smaller the confining pressure and the more obvious the softening property of the material are. Additionally, the lower the gel content and the larger the confining pressure is, the more the hardening property of the material is obvious. Figure 3 mainly intends to show such a dynamic change process.
I have revised the manuscript according to your comments and marked it in the corresponding position.